# Reparameterization through Spatial Gradient Scaling

**Alexander Detkov[1,\*], Mohammad Salameh[2,†], Muhammad Fetrat Qharabagh[1,\*,†], Jialin Zhang[3], Wei Lui[2], Shangling Jui[3], Di Niu[1]**

[1]University of Alberta, [2]Huawei Technologies, [3]Huawei Kirin Solutions
{detkov, fetratqh, dniu}@ualberta.ca
{mohammad.salameh, jui.shangling}@huawei.com
{zhangjialin10, robin.luwei}@hisilicon.com

## Abstract

Reparameterization aims to improve the generalization of deep neural networks by transforming convolutional layers into equivalent multi-branched structures during training. However, there exists a gap in understanding how reparameterization may change and benefit the learning process of neural networks. In this paper, we present a novel spatial gradient scaling method to redistribute learning focus among weights in convolutional networks. We prove that spatial gradient scaling achieves the same learning dynamics as a branched reparameterization yet without introducing structural changes into the network. We further propose an analytical approach that dynamically learns scalings for each convolutional layer based on the spatial characteristics of its input feature map gauged by mutual information. Experiments on CIFAR-10, CIFAR-100, and ImageNet show that without searching for reparameterized structures, our proposed scaling method outperforms the state-of-the-art reparameterization strategies at a lower computational cost. The code is available at https://github.com/Ascend-Research/Reparameterization.

## 1 Introduction

The ever-increasing performance of deep learning is largely attributed to progress made in neural architectural design, with a trend of not only building deeper networks (Krizhevsky et al., 2012; Simonyan & Zisserman, 2014) but also introducing complex blocks through multi-branched structures (Szegedy et al., 2015; 2016; 2017). Recently, efforts have been devoted to Neural Architecture Search, Network Morphism, and Reparametrization, which aim to strike a balance between network expressiveness, performance, and computational cost. Neural Architecture Search (NAS) (Elsken et al., 2018; Zoph & Le, 2017) searches for network topologies in a predefined search space, which often involves multi-branched micro-structures. Examples include the DARTS (Liu et al., 2019) and NAS-Bench-101 (Ying et al., 2019) search spaces that span a large number of cell (block) topologies which are stacked together to form a neural network. In Network Morphism (Wei et al., 2016; 2017), a well-trained parent network is morphed into a child network with the goal of adopting it on a downstream application with minimum re-training. Morphism preserves the parent network's functions and output while yielding child networks that are deeper and wider.

Structural reparameterization (Ding et al., 2021c) attempts to branch and augment certain operations during training into an equivalent but more complex structure with extra learnable parameters. For example, Asymmetric Convolution Block (ACB) (Ding et al., 2019) augments a regular 3x3 convolution with both horizontal $1 \times 3$ and vertical $3 \times 1$ convolutions, such that training is performed on the reparameterized network which takes advantage of the changed learning dynamics. During inference, the trained reparameterized network is equivalently transformed back to its base simple structure, preserving the original low inference time, while maintaining the boosted performance of the reparameterized model. However, there exists a gap regarding the understanding of how and

---

*Work done during an internship at Huawei
†Equal contribution

when the different learning dynamics of a reparameterized model could help its training. In addition, the search for the optimal reparameterized structure over a discrete space (Huang et al., 2022) inevitably increases the computational cost in deep learning.

In this paper, we propose Spatial Gradient Scaling (SGS), an approach that changes learning dynamics as with reparameterization, yet without introducing structural changes to the neural network. We examine the question—*Can we achieve the same effect as branched reparameterization on convolutional networks without changing the network structure?* Our proposed spatial gradient scaling learns a spatially varied scaling for the gradient of convolutional weights, which we prove to have the same learning dynamics as branched reparameterization, without modifying network structure. We further show that scaling gradients by examining the spatial dependence of neighboring pixels in the input (or intermediate) feature maps can boost neural network learning performance, without searching for the reparameterized form. Our main contributions can be summarized as follows:

- We investigate a new problem of spatially scaling gradients in the kernels of convolutional neural networks. This method enhances the performance of existing architectures only by redistributing learning rates spatially, i.e., by adaptively strengthening or weakening the gradients of convolution weights according to their relative position in the kernel.
- We mathematically establish a connection between the proposed SGS and parallel convolutional reparameterization, and show their equivalence in learning. This enables an understanding of how the existing multi-branched reparameterization structures help improve feature learning. This interpretation also suggests an architecture-independent reparameterization method by directly inducing the effect via scaled gradients, bypassing the need for complex structure augmentations, and saving on computational costs.
- We propose a lightweight method to compute gradient scalings for a given network and dataset based on the spatial dependencies in the feature maps. Specifically, we make novel use of the inherent mutual information between neighboring pixels of a feature map within the receptive field of a convolution kernel to dynamically determine gradient scalings for each network layer, with only a minimum overhead to the original training routine.

Extensive experiments show that the proposed data-driven spatial gradient scaling approach leads to results that compete or outperform state-of-the-art methods on several image classification models and datasets, yet with almost no extra computational cost and memory consumption during training that multi-branched reparameterization structures require.

## 2 RELATED WORK

### 2.1 MULTI-BRANCH STRUCTURES

VGG (Simonyan & Zisserman, 2014) is a base model for several computer vision tasks. Due to its limitations, several new structures have been proposed with multiple branches to achieve higher performance. GoogleNet (Szegedy et al., 2015) and Inception (Szegedy et al., 2015; 2016; 2017) architectures deploy multi-branch structures to enrich the learned feature space. ResNet (He et al., 2016) uses a simplified two-branch structure that adds the input of a layer to its output through residual connections. The improvements in top-1 accuracy of ImageNet classification using these structures demonstrate the importance of multiple receptive fields (e.g., $1 \times 1$, $1 \times K$, $K \times 1$, and $K \times K$ convolutions), diverse connections of layers and combination of parallel branches. These performance improvements often come at a computational cost, as complex model topologies are less hardware friendly, and have increased computational requirements. Outside of expertly designed networks, advancements in Neural Architecture Search (NAS) allow for the automation of network design. Several search spaces and discovered high-performing networks (Ying et al., 2019; Dong & Yang, 2020; Ding et al., 2021a) utilize multi-branch structures, which shows their ubiquity in modern convolution architectures. Due to the enormous possibilities of branched model topologies, search is often computationally expensive and requires vast computational resources.

### 2.2 STRUCTURAL REPARAMETRIZATION

Multi-branch structures enhance the performance of ConvNets. This comes at the cost of higher memory and computational power requirements, which is undesirable for inference-time applica-

tions. Structural reparameterization solves this by training with a complex multi-branch model to improve the learned representations but equivalently transforming back to the original simple base model during inference for decreased computational costs. RepVGG (Ding et al., 2021c) introduces a family of VGG-like inference models that are trained with ResNet-inspired reparameterization blocks. These reparameterization blocks consist of parallel $3 \times 3$ and $1 \times 1$ convolutions along with an identity branch. After training, using the linearity of convolutions, parallel branches are equivalently transformed into a single $3 \times 3$ convolution. The inference VGG-like model has the advantage of both the enhanced learned representations of the complex reparameterized model, and the fast and efficient inference of the simple base model. Similarly, DBB (Ding et al., 2021b) structurally reparameterize models by replacing all $K \times K$ convolution with a multi-branch topology composed of multi-scale and sequential $1 \times 1$-$K \times K$ convolutions and average pooling during training. After training, DBB blocks are equivalently reparameterized back to $K \times K$ convolutions for efficient inference. Instead of structurally reparameterizing all convolutions, DyRep (Huang et al., 2022) aims to selectively reparameterize only important convolutions to improve the training efficiency of the reparameterized model. Our spatial gradient scaling approach has the benefit of a branched reparameterization without the added training cost of an augmented network structure.

ACNet (Ding et al., 2019), through pruning experiments, showed that convolution weights on the central crisscross positions of the $3 \times 3$ kernels are more important to the model's representational capacity than corner weights. To further enhance the kernel crisscross's importance, they reparameterize, during training, $3 \times 3$ convolutions with their Asymmetric Convolution Blocks (ACB). ACB comprises of parallel $3 \times 3$, $1 \times 3$, and $3 \times 1$ convolutions. They found that models trained with ACB reparameterization perform better than the base model. Like ACNet, we also emphasize the importance of kernel central positions. However, instead of using ACB blocks, which add significant training cost, we scale the gradients of convolution weights with a spatially varying gradient scaling. In fact, by using spatial gradient scaling, we can emulate the presence of multi-branch topology without adding to the structure and computational cost of the training model.

## 2.3 FEATURE SELECTION WITH MUTUAL INFORMATION

Mutual information has extensive applications in the domain of computer vision and medical imaging. Mutual information is used by (Pluim et al., 2003). Viola & Wells III (1997) as a metric for comparing the alignment of a 3D model to video images. Russakoff et al. (2004) use mutual information to measure similarity between images. In deep learning, Cheng et al. (2018) used the mutual information between inputs, outputs, and target labels of a neural network to infer its power of distinguishing between classes. In this paper, we use mutual information in a novel way to capture dependencies between neighboring elements within a feature map. We use this spatial information as a dynamic scale for adjusting the importance of spatial positions in a convolution kernel.

## 3 METHOD

In this section, we first introduce spatial gradient scaling for convolutional neural networks. We then establish its connection to reparameterization and mathematically show their equivalence. Finally, we describe our mutual-information-based approach to dynamically determine spatial gradient scaling during the training process at a low computational cost.

### 3.1 SPATIAL GRADIENT SCALING

Gradient scaling adjusts backpropagation by strengthening or diminishing gradients of learnable parameters based on the significance of their position in the convolutional kernel. Let $W_l^{(t)}$ be the learnable weights for the $l$-th convolutional layer at training iteration $t$. The shape of $W_l^{(t)}$ is denoted by $(c_{out_l}, c_{in_l}, k_{x_l}, k_{y_l})$, where $c_{out_l}$ and $c_{in_l}$ are sizes of output and input channels, respectively, and $(k_{x_l}, k_{y_l})$ denotes the kernel size. During each iteration of training, we backpropagate the training loss $\mathcal{L}_{train}$ to calculate the gradients of the learnable parameters $\partial \mathcal{L} / \partial W_l^{(t)}$. Following matrix calculus notations, let $\partial \mathcal{L} / \partial W$ denote a derivative tensor whose element at the position indexed by $(m, n, o, p)$ is given by $\partial \mathcal{L} / \partial W_{m,n,o,p}$. Then, a gradient descent optimization step for

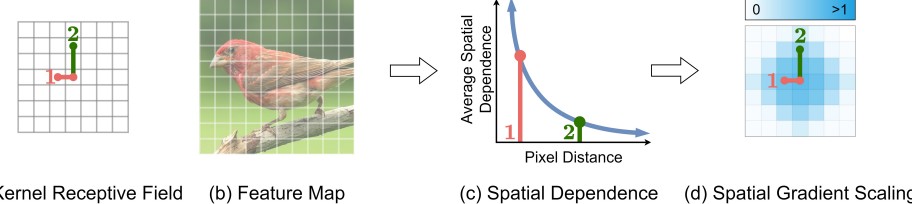

(a) Kernel Receptive Field    (b) Feature Map      (c) Spatial Dependence    (d) Spatial Gradient Scaling

Figure 1: Overview of the framework for learning spatial gradient scalings. In a), we show the kernel receptive field along with elements and their associated pixel distance to the center. We generate a discrete average spatial dependence vs. pixel distance function (c) from the input feature map. Using (c) and pixel distances in (a), we generate the spatial gradient scaling (d). Note that we simplify the process in the figure by considering pixel distance instead of displacement.

the convolutional layer $l$ with weights $W_l$ can be represented by

$$W_l^{(t+1)} \Leftarrow W_l^{(t)} - \lambda^{(t)} \ f\left(\frac{\partial \mathcal{L}}{\partial W_l^{(t)}}, \ W_l^{(t)}, \ ..., \ \frac{\partial \mathcal{L}}{\partial W_l^{(0)}}, \ W_l^{(0)}\right), \tag{1}$$

where $\lambda^{(t)}$ is learning rate and $f(\cdot)$ is an optimizer dependent element-wise function of the current and past gradients and weights. Our approach scales the gradient by a spatial gradient scaling matrix $G_l^{(t)}$, to yield the following gradient descent update rule:

$$W_l^{(t+1)} \Leftarrow W_l^{(t)} - \lambda^{(t)} f\left(G_l^{(t)} \odot \frac{\partial \mathcal{L}}{\partial W_l^{(t)}}, \ W_l^{(t)}, \ ..., \ G_l^{(0)} \odot \frac{\partial \mathcal{L}}{\partial W_l^{(0)}}, \ W_l^{(0)}\right), \tag{2}$$

where $G_l^{(t)}$ is a matrix of shape $(k_{x_l}, k_{y_l})$ and $\odot$ denotes element-wise multiplication along dimensions $c_{out_l}$ and $c_{in_l}$. We additionally constrain $G_l^{(t)}$ to be strictly positive with a mean of 1 to prevent large changes in the overall gradient direction and magnitude. To account for various optimizers, we scale the gradient before any optimizer-dependent calculations like momentum and weight decay.

Elements of $G_l^{(t)}$ are learned in three steps. First, we define each element in $G_l^{(t)}$ by its displacement from the center element. Next, we measure the average spatial relatedness between every two pixels in the input feature map that are particularly displaced apart. We denote this as the average spatial dependence and define it mathematically as a function of the feature map and displacement in Section 3.3. Finally, we assign values to elements in $G_l^{(t)}$ based on the average spatial dependence of their displacement. The overall process is depicted in Figure 1. We assign higher learning priority to elements with higher average spatial dependence. Feature maps with a high spatial correlation over large displacements give rise to more uniform spatial gradient scalings. Feature maps with low spatial correlations yield center concentrated scalings.

## 3.2 EQUIVALENCE TO REPARAMETERIZATION

We now establish the relationship between the proposed spatial gradient scaling and parallel convolution reparameterization. Specifically, we show that backward propagation for a single convolution is different from that of its $N$-branch reparameterization, where the latter is equivalent to updating the original convolution with certain spatial gradient scaling.

Consider a $k_{x_l} \times k_{y_l}$ convolutional layer $l$ with trainable weights $W_l^{(t)} \in \mathbb{R}^{c_{out_l} \times c_{in_l} \times k_{x_l} \times k_{y_l}}$, input $X_l^{(t)}$ and output $Y_l^{(t)}$ at timestep $t$ in training. Following Ding et al. (2019) we can reparameterize this single convolutional layer into a general $N$-branch convolutional structure as depicted in Figure 2 (b) with batchnorms after the reparameterization instead of within. Each branch $n$ contains a convolution with a receptive field no larger than $(k_{x_l}, k_{y_l})$. To mathematically represent variable-sized kernels in different branches, we let $M_{l,n} \in \{0, 1\}^{k_{x_l} \times k_{y_l}}$ denote a binary mask, which is a matrix in the shape of the corresponding kernel's receptive field, as illustrated in Figure 2 (c).

Considering a forward pass, for each original convolutional layer $l$ with weights $W_l^{(t)}$, we can find an equivalent $N$-branch reparameterization, where each branch $n$ contains a convolutional weight

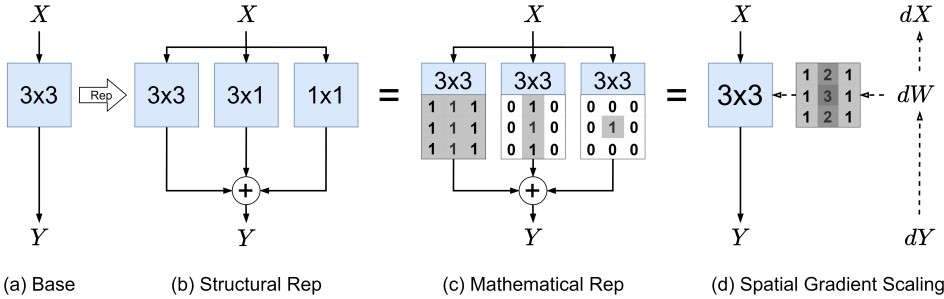

Figure 2: An illustration of the equivalence between SGS and structural reparameterization. The base 3x3 convolution, (a), is structurally reparameterized into (b), a 3-branched reparameterization with diverse receptive fields. (c) shows the equivalent binary masked convolutions. (d) shows the equivalent spatial gradient scaling, which is the sum over the binary masks. The gradient is element-wise multiplied by the spatial gradient scaling before passing to the optimizer.

tensor $W_{l,n}^{(t)}$ and associated binary mask $M_{l,n}$, such that the reparameterization yields the same mapping as the original convolution, i.e.,

$$Y_l^{(t)} = W_l^{(t)} * X_l^{(t)} = \left( \sum_{n=1}^{N_l} M_{l,n} \odot W_{l,n}^{(t)} \right) * X_l^{(t)} = \sum_{n=1}^{N_l} \left( \left( M_{l,n} \odot W_{l,n}^{(t)} \right) * X_l^{(t)} \right), \quad (3)$$

where $\odot$ denotes element-wise multiplication across each dimension $c_{out_l}$ and $c_{in_l}$. The second equality in Eq. 3 decomposes the weight tensor $W_l$, while the third equality achieves branched structural form. The equivalence between a convolutional layer and its $N$-branch reparameterization with masked representation for each branch is illustrated in Figure 2 (c).

However, in backward passes, $W_l$ updates differently in the reparameterized network than in its original form. That is, one step of gradient descent on the reparameterization in Eq. 3 has the following form to yield the merged weights $W_l^{(t+1)}$ for the next timestep:

$$W_l^{(t+1)} \Leftarrow W_l^{(t)} - \lambda^{(t)} \sum_{n=1}^{N_l} \left( M_{l,n} \odot f \left( \frac{\partial \mathcal{L}}{\partial W_{l,n}^{(t)}}, W_{l,n}^{(t)}, ..., \frac{\partial \mathcal{L}}{\partial W_{l,n}^{(0)}}, W_{l,n}^{(0)} \right) \right) \quad (4)$$

Note that the learning dynamics for the reparameterized network differ from the original convolution in Eq. 1, which explains why it may attain higher generalization although having identical expressivity. Despite their different topologies, we show in the following lemma that updates for the original convolution (Eq. 1) and any of its $N$-branch reparameterization (Eq. 4) differ only by a constant spatial gradient scaling $G_l$.

**Lemma 1.** *Assume $f(\cdot)$ is a linear function in a gradient descent optimization algorithm (e.g., momentum, weight decay). For any reparameterization of a convolutional layer $l$ that can be represented as a summation of $N$ convolutional branches with weights $W_{l,n}^{(t)}$ and binary receptive field mask $M_{l,n}$, for $n = 1, \ldots, N$, its gradient descent update Equation 4 is equivalent to*

$$W_l^{(t+1)} \Leftarrow W_l^{(t)} - \lambda^{(t)} f \left( G_l \odot \frac{\partial \mathcal{L}}{\partial W_l^{(t)}}, W_l^{(t)}, ..., G_l \odot \frac{\partial \mathcal{L}}{\partial W_l^{(0)}}, W_l^{(0)} \right) \quad (5)$$

*where $G_l = \sum_{n=1}^{N} M_{l,n}$ is a spatial gradient scaling applied to the original convolution.*

The proof in Appendix A follows readily from the equations of gradient descent and the linearity of convolutions. Lemma 1 has multiple implications. First, it provides an understanding of how branched reparameterization helps to change the backpropagation dynamics; it redistributes learning rates spatially to focus on more important weights in a convolutional kernel. Second, Lemma 1 allows us to convert the search for a reparameterization structure into an equivalent numerical gradient scaling search, which is more efficient and lends itself to analytical methods (as we demonstrate

in Section 3.3). Our scaling interpretation of structural reparameterization allows us a more flexible search space unconstrained by the computational complexity of the underlying structure. Third, we find agreement between our formalized gradient scaling approach and observational rules of thumb in the reparameterization literature. For example, the current trend of preferring branches with a diverse range of convolutional receptive fields presented by Ding et al. (2021b) may stem from the fact that without it, the gradient scaling is uniform and loses its spatial emphasis.

## 3.3 MUTUAL INFORMATION BASED SPATIAL GRADIENT SCALING

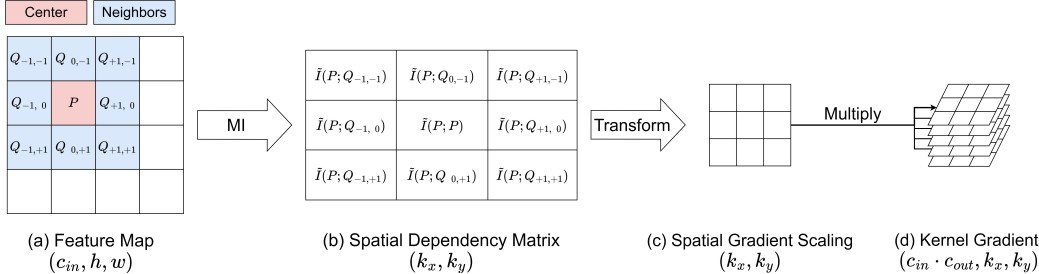

Figure 3: Overview of SGS calculation from the spatial dependencies in the input feature map. (a) a joint distribution is estimated for every pixel in the feature map and its $(i, j)$ neighbor denoted by random variables $P$ and $Q_{i,j}$ respectively (b) Mutual information is calculated between $P$ and $Q_{i,j}$ for all $(i, j)$ and values are placed in spatial dependence matrix $(S_l)$ (c) $S_l$ is transformed to SGS $(G_l)$ by equation 8 (d) $G_l$ is element-wise multiplied to the kernel gradients.

We describe our approach to finding spatial gradient scalings from the spatial dependencies in the input feature map. We begin by showing how mutual information can be used to quantify the average spatial dependency between $(i, j)$-displaced pixels. We then assign values to elements of our gradient scalings based on their spatial displacement to the center of the kernel and its associated average spatial dependency. Figure 3 shows an overview of the method.

We quantify the spatial dependence of two pixels through the use of probabilistic dependence. We define random variables $P_l$ and $Q_{l,i,j}$ as pixel values in the $l$-th layer feature map and their associated $(i, j)$-displaced neighbor values, respectively. The $(i, j)$-displaced neighbor for a pixel is the corresponding pixel $(i, j)$ units away. We express spatial dependence, $S_l(i, j)$, as the normalized mutual information (MI), $\tilde{I}(\cdot; \cdot)$ of the random variables $P_l$ and $Q_{l,i,j}$:

$$S_l(i, j) = \tilde{I}(P_l; Q_{l,i,j}), \tag{6}$$

Displacement $(i, j)$ spans the entire $(k_{x_l}, k_{y_l})$ receptive field of the convolution kernel as demonstrated in Figure 3 (a). We arrange elements of $S_l(i, j)$ into a spatial dependence matrix as illustrated in Figure 3 (b). Intuitively, spatial dependency within pixels results in statistical dependencies in pixel values, which can be detected via mutual information. $\tilde{I}(\cdot)$ is bounded between 0, when variables are independent, and 1, with complete mutual dependence.

We calculate the normalized MI from the Shannon entropy $H(\cdot)$ of the random variables:

$$\tilde{I}(P_l, Q_{l,i,j}) = \frac{H(P_l) + H(Q_{l,i,j}) - H(P_l, Q_{l,i,j})}{H(P_l, Q_{l,i,j})} \tag{7}$$

where $H(P_l, Q_{l,i,j})$ is the joint entropy of $P$ and $Q_{i,j}$. We calculate the entropy by estimating the distributions of $P_l$ and $Q_{l,i,j}$ through discrete binning. High-resolution images often contain redundant / spatially repeated neighbor pixels that may cause mutual information to overestimate the amount of useful learnable spatial dependence. To account for this, on ImageNet, we remove occurrences of $P_l$ with $Q_{l,i,j}$ that are near in pixel values when we estimate their distributions. In practice, a single image is not large enough to generate accurate distributions for $P_l$ and $Q_{l,i,j}$, so we aggregate pixel and their $(i, j)$-neighbors values over multiple batches of training data. Due to the inexpensive nature of the mutual information computation, we can afford to perform this calculation for each convolutional layer and every couple of epochs.

| Dataset | Rep Method | Cost (GPU hrs.) | FLOPs (M) | Params (M) | Acc (%) |
|---------|-----------|-----------------|-----------|------------|---------|
| CIFAR-10 | Origin | 2.3 | 313 | 15 | $94.90_{\pm0.07}$ |
| | DBB* | 9.4 | 728 | 34.7 | $94.97_{\pm0.06}$ |
| | DBB | 7.0 | 728 | 34.7 | $95.00_{\pm0.1}$ |
| | DyRep* | 6.9 | 597 | 26.4 | $95.22_{\pm0.13}$ |
| | DyRep | 7.2 | 674 | 25.5 | $95.00_{\pm0.06}$ |
| | SGS (ours) | 2.5 | 313 | 15 | $95.20_{\pm0.05}$ |
| CIFAR-100 | Origin | 2.3 | 314 | 15 | $73.70_{\pm0.1}$ |
| | DBB* | 9.4 | 728 | 34.7 | $74.04_{\pm0.08}$ |
| | DBB | 7.0 | 728 | 34.7 | $74.40_{\pm0.1}$ |
| | DyRep* | 6.7 | 582 | 27.1 | $74.37_{\pm0.11}$ |
| | DyRep | 6.4 | 560 | 22.8 | $74.10_{\pm0.2}$ |
| | SGS (ours) | 2.5 | 313 | 15 | $75.50_{\pm0.1}$ |

Table 1: Results for VGG-16 on CIFAR-10 and CIFAR-100 trained using the official implementation of DyRep (Huang et al., 2022). Training is done on a single NVIDIA Tesla V100 GPU. FLOPs and Parameters are averaged across DyRep runs. Results marked with * are taken from the official DyRep paper, while the rest are our runs averaged over 5 independent replicas.

To get the spatial gradient scaling, $G_l$, we transform the spatial dependency matrix $S_l$ with an element-wise transform parameterized by a hyperparameter $k$:

$$G_l = \frac{k \times S_l}{(k-1)S_l + 1} \, , \tag{8}$$

$k$ converts mutual information values into effective gradient scalings. Finally, we normalize the mean value of $G_l$. An overview of the SGS framework is given as pseudo-code in Appendix A.5, and details can be found in the corresponding open-source code [1].

In short, our scaling defines how significant weight elements are for feature extraction based on their spatial location within the kernel. In order to determine the scaling, we use mutual information to measure the notion of spatial dependence between pixels a distance apart. We give high learning priority to the elements with a large spatial dependence on the center kernel element.

## 4 EXPERIMENTS AND RESULTS

In this section, we assess the effectiveness of spatial gradient scaling in improving model generalization ability. Following convention (Huang et al. (2022), Ding et al. (2021b)), we compare test accuracies for ResNet and VGG models trained under state-of-the-art reparameterization schemes. We adopt the model training code and strategy from Huang et al. (2022).

### 4.1 CIFAR

We train VGG-16 on CIFAR-{10,100} for 600 epochs with a batch size of 128, cosine annealing scheduler with an initial learning rate of 0.1, and SGD optimizer with momentum 0.9 and weight decay $1 \times 10^{-4}$. We update our spatial gradient scalings every 30 epochs using 20 random batches from the training set. We add a 1 epoch warm-up period at the start of training before generating our gradient scalings. Results are shown in Table 3. Additional results are available in Appendix A.1.

Our framework uses a single hyperparameter $k$, which defines a functional mapping between mutual information and spatial gradient scaling. We search for $k$ on CIFAR100 and use the optimal for experiments on CIFAR10 and ImageNet. We perform a grid search on CIFAR100 and VGG-16 over $k \in \{2, 3, 4, 5, 6, 7\}$ using 20% of the training set for validation. As $k$ acts to interpret values of mutual information into meaningful gradient scalings, we may expect $k$ to remain constant across models and datasets (we test this claim in Section 4.3).

---

[1]https://github.com/Ascend-Research/Reparameterization

| Model | Rep method | Cost (GPU days) | Avg. FLOPs (G) | Avg. params (M) | Acc (%) |
|---|---|---|---|---|---|
| ResNet-18 | Origin | 4.8 | 1.81 | 11.7 | $71.13_{\pm 0.04}$ |
| | DBB* | 8.1 | 4.13 | 26.3 | 70.99 |
| | DyRep* | 6.3 | 2.42 | 16.9 | 71.58 |
| | DyRep | 9.1 | 2.92 | 22.1 | $71.50_{\pm 0.03}$ |
| | SGS (ours) | 5.1 | 1.81 | 11.7 | $71.65_{\pm 0.05}$ |
| ResNet-34 | Origin | 5.3 | 3.66 | 21.8 | $74.17_{\pm 0.05}$ |
| | DBB* | 12.8 | 8.44 | 49.9 | 74.33 |
| | DyRep* | 7.7 | 4.72 | 33.1 | 74.68 |
| | DyRep | 10.6 | 4.95 | 38.3 | $74.40_{\pm 0.03}$ |
| | SGS (ours) | 5.8 | 3.66 | 21.8 | $74.62_{\pm 0.05}$ |
| ResNet-50 | Origin | 7.5 | 4.09 | 25.6 | $76.95_{\pm 0.05}$ |
| | DBB* | 13.7 | 6.79 | 40.7 | 76.71 |
| | DyRep* | 8.5 | 5.05 | 31.5 | 77.08 |
| | DyRep | 11.0 | 5.84 | 38.3 | $77.11_{\pm 0.03}$ |
| | SGS (ours) | 7.9 | 4.09 | 25.6 | $77.10_{\pm 0.01}$ |

Table 2: Results on ImageNet dataset. We use the official implementation of DyRep (Huang et al., 2022) on 8 NVIDIA Tesla V100 GPUs. FLOPs and Parameters are averaged across DyRep runs. Results marked with * are taken from DyRep paper; the rest are our runs averaged over 3 seeds.

Our spatial gradient scaling performs equally or better than state-of-the-art reparameterization methods at a fraction of their cost. On CIFAR10, our spatial gradient scaling outperforms DBB and performs as well as DyRep while only requiring a third of their training time. On CIFAR100, we obtain over $1\%$ accuracy improvement from DBB and DyRep while taking less than half their GPU hours. We attribute the success of spatial gradient scaling to its enhanced reparameterization space and strategy. Unlike structural reparameterization methods like DBB and DyRep, we are not limited by the computational complexity of our blocks, which enables us to explore a much larger space of reparameterizations. Our formalism of spatial gradient scaling as an equivalent to reparameterization also enables us to perform a search in an easily implementable continuous space as opposed to a discrete structural one. Our mutual information strategy adaptively reparameterizes each convolution throughout the training process efficiently and effectively (as we show in Section 4.3).

## 4.2 IMAGENET

We train the ResNet models for 120 epochs, with a batch size of 256, cosine annealing scheduler with initial lr of 0.1, color jitter augmentation, and SGD with a momentum of 0.1 and weight decay $1 \times 10^{-4}$. Scalings update every 5 epochs using two random training batches after a one-epoch model warm-up. Hyperparameter $k$ is taken as the optimal found from the CIFAR100 grid search. Results are presented in Table 2.

As with CIFAR, we see significantly reduced training times with our spatial gradient scaling method for equal or better accuracy compared to the state-of-the-art reparameterization methods. The benefits of reparameterization are attained without complicating the model structure with expensive reparameterization blocks. Large convolution kernels, like the $7 \times 7$ used in ResNet, are difficult to structurally reparameterize. First, these large convolutions are expensive in terms of compute and memory. The addition of a parallel reparameterization branch only increases its computational cost further. Second, as convolution size increases, so does the number of possible diverse receptive fields (for example, $7 \times 7$'s receptive fields are: $1 \times 1, 1 \times 3, ..., 3 \times 5, ...$). DBB and DyRep can only consider a tiny fraction of the possible set ($7 \times 7, 1 \times 7, 7 \times 1, 1 \times 1$). Our spatial gradient scaling can consider all possible receptive fields, even non-standard ones, through our general binary masks. In addition, our method completely avoids the computational pitfalls of structural reparameterization, as our reparameterization happens efficiently on the gradient level.

| Dataset | SGS Method | Cost (GPU hrs.) | Acc (%) |
|---|---|---|---|
| CIFAR-100 | Origin | 0.40 | $69.6_{\pm 0.1}$ |
| | Grid Search | 19.6 | $71.0_{\pm 0.1}$ |
| | Autocorrelation | 0.42 | $71.1_{\pm 0.2}$ |
| | SGS (ours) | 0.42 | $71.5_{\pm 0.1}$ |

Table 3: Results of VGG-11 on CIFAR-100 using different spatial gradient scaling search methods.

## 4.3 ABLATION STUDIES

**Effectiveness of Mutual Information Approach.** We demonstrate our mutual information approach's efficacy in finding high-performance spatial gradient scalings. We compare to autocorrelation, another commonly used dependency measure, as well as a grid search for gradient scalings.

We train all methods on CIFAR100 with VGG-11 for 200 epochs with a batch size of 512, cosine annealing scheduler with an initial learning rate of 0.1, and SGD optimizer with momentum 0.9 and weight decay $5 \times 10^{-4}$. For autocorrelation and mutual information, we update our spatial gradient scalings every 10 epochs using 2 random batches from the training set and a warmup of 1 epoch.

Similar to mutual information, we can measure spatial dependencies using autocorrelation. Specifically, we calculate the correlation of a feature map with itself shifted by $(i, j)$, where $(i, j)$ indexes into a spatial dependency matrix (Figure 3.3). We use the same $k$ transform to map autocorrelation values into gradient scalings. We perform a grid search on over $k \in \{1, 2, 3, 4, 5, 6\}$ using 20% of the training set for validation. For mutual information, we use the optimal $k$ found in Section 4. We take several considerations for grid search over spatial gradient scaling to make the search tractable. First, we reduce the search space by considering a single $3 \times 3$ gradient scaling shared by all convolutional layers and constant for all training epochs. We additionally parameterize the scaling matrix by two variables, $\alpha$, and $\beta$, which determine the ratio of the center element scaling to the edges and corners respectively (shown in Appendix 7). We search over $\alpha, \beta \in \{0.8, 1.0, 1.25, 1.7, 5.0, 10, 100\}$.

Our mutual information approach (SGS) outperforms both autocorrelation and grid search. While grid search is robust, it suffers from high computational complexity, which requires designing a constrained search space. Unlike SGS, grid search cannot effectively adapt across convolution depth and time without an exponential blowup in the search space. While autocorrelation outperforms grid search, our mutual information method performs better. We attribute this to the fact that autocorrelation can only measure linear relationships in the feature map, while mutual information can measure both linear and non-linear dependencies.

**k-Transformation Search.** In this section, we investigate the behavior of the $k$ hyperparameter across models and datasets. Additionally, we corroborate our decision in Section 4 to learn $k$ once on CIFAR-100 and transfer to ImageNet. Following the training strategy defined in Section 4, we train and evaluate models over a range of $k$ values and plot the results in Appendix Figure 8.

We observe that $k$ curves across models and datasets peak near $k = 5$. This may imply that $k = 5$ is a robust default value. We also find consistent performance gains over baseline for a large range of $k$ values. This implies that the conventional uniform update of weights is suboptimal, and lower testing error can be attained via spatial gradient scaling.

## 5 CONCLUSION

In this paper, we present Spatial Gradient Scaling (SGS), an approach that improves the generalization of neural networks by changing the learning dynamics to focus on spatially important weights. We achieve this by scaling the convolutions gradients adaptively from the spatial dependencies of feature maps. We propose a mutual information-based approach to compute the gradient scaling with minimum overhead to the original training routine. We prove that our SGS is equivalent to convolutional reparameterization under certain conditions. This enables us to take advantage of the benefits of reparameterization without introducing complex branching into model structures. Experiments show that our method outperforms the state-of-the-art structural reparameterization approaches on several image classification models and datasets at a much lower computational cost.

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

# A APPENDIX

## A.1 EXPERIMENTS ON CIFAR-100

We report in Table 4 results for the CIFAR-100 dataset on a wide range of CNN architectures. We train for 200 epochs with a batch size of 128, a cosine annealing scheduler with an initial learning rate of 0.1, and an SGD optimizer with momentum 0.9 and weight decay $5 \times 10^{-4}$. For data augmentation, we use random crop, flip, and cutout. Spatial gradient scalings are updated with mutual information every 5 epochs, using 2 random batches from the training set, and with $k = 5$. At the start of training, we add a warm-up period of 1 epoch before generating our gradient scalings. Adding spatial gradient scaling to optimization yields performance gains for several architectures.

| Model | Baseline | SGS (Ours) |
|---|---|---|
| VGG-11 | $70.8_{\pm 0.1}$ | $73.2_{\pm 0.1}$ |
| VGG-13 | $74.7_{\pm 0.1}$ | $76.1_{\pm 0.1}$ |
| VGG-16 | $74.9_{\pm 0.1}$ | $76.2_{\pm 0.1}$ |
| VGG-19 | $74.6_{\pm 0.1}$ | $76.1_{\pm 0.2}$ |
| ResNet-18 | $77.8_{\pm 0.1}$ | $78.8_{\pm 0.1}$ |
| ResNet-34 | $79.0_{\pm 0.3}$ | $80.0_{\pm 0.2}$ |
| ResNet-50 | $79.1_{\pm 0.2}$ | $79.6_{\pm 0.2}$ |
| ShuffleNet | $72.4_{\pm 0.1}$ | $72.8_{\pm 0.2}$ |
| MobileNetV2 | $73.9_{\pm 0.1}$ | $74.2_{\pm 0.2}$ |
| SeResNet-18 | $78.4_{\pm 0.1}$ | $79.6_{\pm 0.1}$ |
| SeResNet-34 | $79.3_{\pm 0.2}$ | $80.3_{\pm 0.2}$ |
| PreActResNet-18 | $75.7_{\pm 0.1}$ | $76.3_{\pm 0.1}$ |
| PreActResNet-34 | $77.4_{\pm 0.1}$ | $77.9_{\pm 0.1}$ |
| PyramidNet ($\alpha = 48$) | $79.1_{\pm 0.1}$ | $79.9_{\pm 0.2}$ |
| PyramidNet ($\alpha = 84$) | $81.1_{\pm 0.1}$ | $81.6_{\pm 0.1}$ |
| PyramidNet ($\alpha = 270$) | $83.3_{\pm 0.1}$ | $83.6_{\pm 0.1}$ |
| Xception | $79.0_{\pm 0.1}$ | $80.0_{\pm 0.1}$ |

Table 4: Results for models trained on CIFAR-100 with and without spatial gradient scaling. Results are averaged over 3 independent replicas.

## A.2 COMPARISONS TO ADAPTIVE GRADIENT OPTIMIZERS

In this section, we compare spatial gradient scaling to popular adaptive gradient optimizers. Unlike adaptive optimizers, which typically optimize based on the model weights and gradients of previous timesteps, our gradient scaling uses the spatial properties within the training data to effectively scale convolution gradients. In Table 5, we present CIFAR-100 results for various optimizers with and without spatial gradient scaling. We tune optimizer hyperparameters with a grid search using 20% of the training data for validation. We focus our search on learning rate and weight decay, leaving other optimizer settings as PyTorch defaults. Training and SGS settings, outside of searched optimizer hyperparameters, are identical to those described in Appendix A.1. We find that spatial gradient scaling improves the performance of even the highest-performing optimizer, SGD + Momentum. Moreover, all tested optimizers, Adagrad being the only exception, benefitted from spatial gradient scaling. Even in Adagrad's case, we can find performance gain by applying gradient scaling post-optimizer calculations and right before the weight update step (shown as Adagrad*) as opposed to directly to the back-propagated gradient and before optimizer calculations.

## A.3 SENSITIVITY TO DIFFERENT TRAINING SETTINGS

In this section, we empirically study performance improvement by spatial gradient scaling across training hyperparameters. Results are shown in Table 6. Training and SGS settings, outside of those in the ablation study, are identical to Appendix A.1. We find performance improvements over the baseline under all tested hyperparameter configurations. This suggests that SGS can be used effectively to improve ConvNets training without requiring extensive hyperparameter tweaking.

| Model | Optimizer | No SGS | SGS (Ours) |
|---|---|---|---|
| VGG-11 | SGD | $68.3_{\pm 0.1}$ | $70.7_{\pm 0.1}$ |
| | SGD + Momentum | $71.0_{\pm 0.1}$ | $73.2_{\pm 0.1}$ |
| | Adadelta | $69.2_{\pm 0.2}$ | $70.8_{\pm 0.1}$ |
| | Adagrad | $66.2_{\pm 0.1}$ | $66.2_{\pm 0.2}$ |
| | Adagrad* | $66.2_{\pm 0.1}$ | $67.0_{\pm 0.1}$ |
| | Adam | $68.8_{\pm 0.2}$ | $70.4_{\pm 0.2}$ |
| | Adamax | $68.4_{\pm 0.2}$ | $69.9_{\pm 0.1}$ |
| | NAdam | $69.2_{\pm 0.1}$ | $70.2_{\pm 0.1}$ |
| | RAdam | $69.8_{\pm 0.1}$ | $71.1_{\pm 0.1}$ |
| ResNet-18 | SGD | $75.8_{\pm 0.2}$ | $76.3_{\pm 0.1}$ |
| | SGD + Momentum | $77.6_{\pm 0.1}$ | $78.5_{\pm 0.1}$ |
| | Adadelta | $75.8_{\pm 0.2}$ | $76.4_{\pm 0.1}$ |
| | Adagrad | $72.2_{\pm 0.2}$ | $72.3_{\pm 0.2}$ |
| | Adagrad* | $72.2_{\pm 0.2}$ | $72.9_{\pm 0.1}$ |
| | Adam | $75.1_{\pm 0.1}$ | $76.2_{\pm 0.1}$ |
| | Adamax | $74.9_{\pm 0.1}$ | $75.8_{\pm 0.1}$ |
| | NAdam | $75.6_{\pm 0.1}$ | $76.0_{\pm 0.1}$ |
| | RAdam | $76.2_{\pm 0.2}$ | $76.8_{\pm 0.1}$ |

Table 5: Results for models trained on CIFAR-100 using a range of optimizers, with and without spatial gradient scaling. Results are averaged over 3 independent replicas.

| Model | Epochs | Batch Size | Baseline | SGS (Ours) |
|---|---|---|---|---|
| VGG-11 | 100 | 64 | $70.4_{\pm 0.2}$ | $71.1_{\pm 0.1}$ |
| | | 128 | $70.8_{\pm 0.1}$ | $72.5_{\pm 0.2}$ |
| | | 512 | $68.8_{\pm 0.2}$ | $70.8_{\pm 0.1}$ |
| | 200 | 64 | $71.0_{\pm 0.1}$ | $72.4_{\pm 0.2}$ |
| | | 128 | $70.8_{\pm 0.1}$ | $73.2_{\pm 0.1}$ |
| | | 512 | $68.6_{\pm 0.2}$ | $71.6_{\pm 0.1}$ |
| | 600 | 64 | $71.0_{\pm 0.1}$ | $73.0_{\pm 0.1}$ |
| | | 128 | $70.1_{\pm 0.2}$ | $73.2_{\pm 0.1}$ |
| | | 512 | $68.2_{\pm 0.1}$ | $71.3_{\pm 0.1}$ |
| ResNet-18 | 100 | 64 | $77.5_{\pm 0.2}$ | $78.1_{\pm 0.1}$ |
| | | 128 | $77.5_{\pm 0.1}$ | $78.0_{\pm 0.2}$ |
| | | 512 | $75.2_{\pm 0.1}$ | $75.6_{\pm 0.1}$ |
| | 200 | 64 | $77.8_{\pm 0.2}$ | $78.4_{\pm 0.1}$ |
| | | 128 | $77.8_{\pm 0.1}$ | $78.8_{\pm 0.1}$ |
| | | 512 | $76.2_{\pm 0.1}$ | $76.7_{\pm 0.1}$ |
| | 600 | 64 | $78.0_{\pm 0.1}$ | $78.7_{\pm 0.1}$ |
| | | 128 | $77.9_{\pm 0.1}$ | $79.0_{\pm 0.1}$ |
| | | 512 | $76.4_{\pm 0.1}$ | $77.2_{\pm 0.2}$ |

Table 6: Performance gain of spatial gradient scaling on CIFAR-100 using a range of training hyperparameters with and without spatial gradient scaling. Results are averaged over 3 independent replicas.

## A.4 Experiments on Semantic Segmentation and Human Pose Estimation

We consider semantic segmentation and human pose estimation to evaluate our spatial gradient scaling. For semantic segmentation, we adapt our training code and strategy from SemSeg [2]. We train and evaluate the PSPNet models on the Cityscapes dataset. PSPNet is composed of a ResNet-18/34/50 backbone and an up-sampling head for pixel-level classification. Using the backbone weights of the ResNet models pre-trained on ImageNet with and without SGS (Section 4.2), we fine-tune with and without SGS on Cityscapes. Results are shown in Table 7.

| Model | Baseline (mIoU/mAcc/allAcc) | SGS (mIoU/mAcc/allAcc) |
|---|---|---|
| PSPNet-18 | 72.8/80.0/95.3 | 73.7/81.1/95.4 |
| PSPNet-34 | 76.1/84.0/95.7 | 76.8/84.3/95.9 |
| PSPNet-50 | 75.4/82.8/95.8 | 75.6/82.8/95.9 |

Table 7: Results for PSPNet models trained and fine-tuned with and without spatial gradient scaling on the semantic segmentation task.

For human pose estimation, we use the training code and strategy from Pytorch-Pose-HG-3D [3]. We train and evaluate on MPII dataset with MSRA ResNet. MSRA ResNet has a Resnet-18/34/50 backbone which we take pretrained from ImageNet with and without SGS. We then fine-tune with and without SGS on MPII. Results are shown in Table 8.

| Model | Baseline (PCKh@0.5) | SGS (PCKh@0.5) |
|---|---|---|
| MSRA ResNet-18 | 83.8 | 84.4 |
| MSRA ResNet-34 | 85.8 | 86.2 |
| MSRA ResNet-50 | 85.3 | 85.6 |

Table 8: Results for MSRA ResNet models trained and fine-tuned with and without spatial gradient scaling on the human pose estimation task.

## A.5 Training Pseudo Code

---

**Algorithm 1** Spatial Gradient Scaling Training

---

**Input:** Training dataset $X$. Set of spatial gradient scaling $SGS$. Number of epochs $E$ for training. Number of SGS warmup epochs $N_e$. Number of batches $N_b$ for SGS calculation. Set of convolution layers $L$.

    **for** $e = 0, ..., (E-1)$ **do**
        **if** $e$ divisible by $N_e$ **then**
            $b \leftarrow$ randomly sample $N_b$ batches from $X$
            Forward propagate $b$ and assign the input feature map of convolution layer $l$ to the set $X_l$
            **for** $l$ in $L$ **do**
                **for** $(i, j)$ in kernel size of $l$ **do**
                    $P, Q_{(i,j)} \leftarrow$ List of pixels of $X_l$ and their corresponding (i,j)th neighbours
                    $SGS_{l,(i,j)} \leftarrow$ Mutual Information between $P$ and $(i, j)$ neighbour $Q_{(i,j)}$
        **for** $(x, y)$ in $(X, Y)$ **do**
            Forward and backward propagate $(x, y)$
            **for** $l$ in $L$ **do**
                **for** $(i, j)$ in kernel size of $l$ **do**
                    Scale the $l$th convolution spatial $(i, j)$ gradient element with $SGS_{l,(i,j)}$
        Update weights with gradients

---

---
[2] https://github.com/hszhao/semseg
[3] https://github.com/xingyizhou/pytorch-pose-hg-3d

## A.6 LEARNED SPATIAL GRADIENT SCALING

In Figure 4, we plot the spatial gradient scalings for the first, seventh, and last convolutional layers of VGG-16 at the start and the end of its training on CIFAR-100. We observe more uniformly distributed gradient scaling for beginning layers giving equal relative importance to all kernel weights. Deeper layers have center-focused gradient scalings with most of the importance distributed on the center and edge elements as opposed to the corners. Interestingly, we observe that after training, spatial gradient scalings for deeper layers become even more center-focused, indicating a decrease in the spatial dependencies of the feature map.

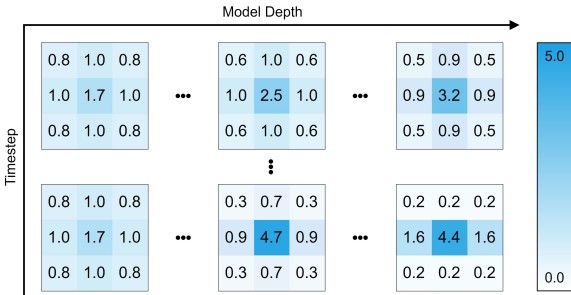

Figure 4: Spatial gradient scaling for the first, seventh, and last convolutional layer of VGG16 on CIFAR-100 at the beginning and end of training.

We plot gradient scalings for ResNet18 on ImageNet in Figure 5. Like CIFAR, we see uniform spatial gradient scalings in early layers and center-focused scaling in deeper layers. Contrary to CIFAR, however, we find that gradient scalings "smoothen" over time and become more uniform. Discrepancies between the behavior of spatial gradient scaling on CIFAR and ImageNet warrant future investigation.

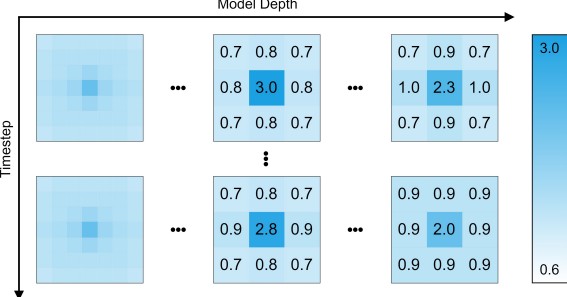

Figure 5: Spatial gradient scaling for the first, eight, and last convolutional layers of ResNet18 on ImageNet at the beginning and end of training. The first layer is a $7 \times 7$ convolution.

## A.7 WEIGHT MAGNITUDES

Similar to Ding et al. (2019), we investigate the average kernel magnitude matrix of learned weights with and without our spatial gradient scaling. For a convolution weight, the average kernel magnitude matrix is defined as the mean of the absolute value of the weight tensor across the input and output channels (leaving the spatial channels intact). We further normalize the mean of the matrix for meaningful comparisons.

Figure 6 depicts the average kernel magnitude matrix for the first, seventh, and last convolutional layer of VGG-16 trained on the CIFAR-100 dataset with and without spatial gradient scaling. We additionally show the spatial gradient scaling of the last training epoch of the SGS training scheme. Similar to Ding et al. (2019), we observe that our spatial gradient scaling modifies the normally

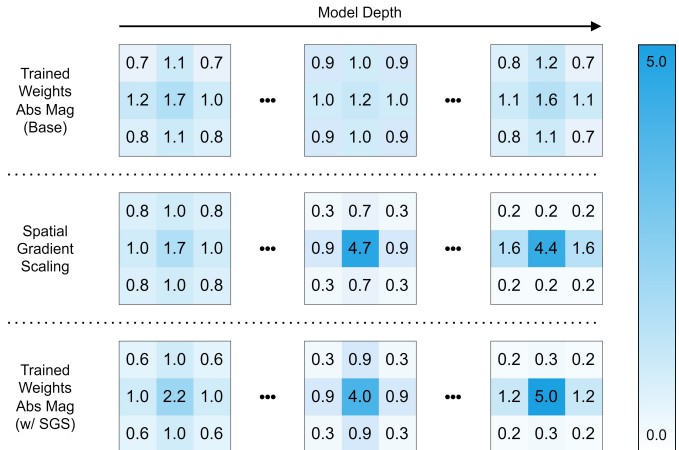

Figure 6: Comparison of the average kernel magnitude matrix of the first, seventh, and last convolution weight between regularly trained VGG16 on CIFAR-100 and the network trained with gradient scaling. We also show the spatial gradient scaling during the last epoch of SGS training.

trained kernel magnitude matrix to focus more on the center and edge elements. We also qualitatively observe the similarities between the shape of spatial gradient scaling and the final trained weights.

## A.8 PROOF FOR EQUIVALENCE TO REPARAMETERIZATION

In this section we prove by induction lemma 1 for optimizers that are linear functions of current and past weights and gradients i.e.,

$$
f\left(\frac{\partial \mathcal{L}}{\partial W_l^{(t)}}, W_l^{(t)}, ..., \frac{\partial \mathcal{L}}{\partial W_l^{(0)}}, W_l^{(0)}\right) = \sum_{\tau=0}^{t}\left(\gamma^{(\tau)}\frac{\partial \mathcal{L}}{\partial W_l^{(\tau)}} + \zeta^{(\tau)}W_l^{(\tau)}\right)
$$

where $W_l^{(t)}$ and $\partial \mathcal{L}/\partial W_l^{(t)}$ are the weights and respective gradients for the $l$-th convolutional layer at training iteration $t$ and $f(\cdot)$ is a linear optimizer parameterized by arbitrary $\gamma^{(\tau)}$ and $\zeta^{(\tau)}$.

We begin by defining two models: a base convolutional layer, and its n-branch reparameterization which at $t = 0$ has an identical mapping:

$$
Y_l = W_l^{(0)} * X_l = \left(\sum_{n=1}^{N} M_{l,n} \odot W_{l,n}^{(0)}\right) * X_l = \sum_{n=1}^{N}\left(\left(M_{l,n} \odot W_{l,n}^{(0)}\right) * X_l\right)
$$

Our goal is to find a modified update rule for the single convolution such the mappings are identical throughout training i.e., $\forall(t \geq 0)$ the following holds:

$$
W_l^{(t)} = \sum_{n=1}^{N} M_{l,n} \odot W_{l,n}^{(t)} \tag{9}
$$

Assume equation 9 holds $\forall(t \leq t_0)$ (we know this is for sure the case for $t_0 = 0$). We wish to find an update rule for the single convolution such that equation 9 holds for $t = t_0 + 1$ and thus $\forall(t \leq (t_0 + 1))$. Together with our assumption of identical mapping at $t = 0$, we can then ensure equation 9 holds $\forall(t \geq 0)$.

Assuming equation 9 holds $\forall (t \leq t_0)$ we find the update equation for the merged weights $W_l^{t_0+1}$ as:

$$W_l^{(t_0+1)} \Leftarrow \sum_{n=1}^{N} M_{l,n} \odot \left( W_{n,l}^{(t_0)} - \lambda^{(t_0)} \sum_{\tau=0}^{t_0} \left( \gamma^{(\tau)} \frac{\partial \mathcal{L}}{\partial W_{l,n}^{(\tau)}} + \zeta^{(\tau)} W_{l,n}^{(t_0)} \right) \right),$$

$$= \sum_{n=1}^{N} \left( M_{l,n} \odot W_{l,n}^{(t_0)} \right) - \lambda^{(t_0)} \sum_{n=1}^{N} \left( M_{l,n} \odot \sum_{\tau=0}^{t_0} \left( \gamma^{(\tau)} \frac{\partial \mathcal{L}}{\partial W_{l,n}^{(\tau)}} + \zeta^{(\tau)} W_{l,n}^{(\tau)} \right) \right)$$

$$= W_l^{(t_0)} - \lambda^{(t_0)} \sum_{n=1}^{N} \left( M_{l,n} \odot \sum_{\tau=0}^{t_0} \left( \gamma^{(\tau)} \frac{\partial \mathcal{L}}{\partial W_{l,n}^{(\tau)}} + \zeta^{(\tau)} W_{l,n}^{(\tau)} \right) \right) \qquad \text{Equation 9}$$

$$= W_l^{(t_0)} - \lambda^{(t_0)} \sum_{n=1}^{N} \left( \sum_{\tau=0}^{t_0} \left( \gamma^{(\tau)} M_{l,n} \odot \frac{\partial \mathcal{L}}{\partial W_{l,n}^{(\tau)}} + \zeta^{(\tau)} M_{l,n} \odot W_{l,n}^{(\tau)} \right) \right)$$

$$= W_l^{(t_0)} - \lambda^{(t_0)} \sum_{\tau=0}^{t_0} \left( \sum_{n=1}^{N} \left( \gamma^{(\tau)} M_{l,n} \odot \frac{\partial \mathcal{L}}{\partial W_{l,n}^{(\tau)}} + \zeta^{(\tau)} M_{l,n} \odot W_{l,n}^{(\tau)} \right) \right)$$

$$= W_l^{(t_0)} - \lambda^{(t_0)} \sum_{\tau=0}^{t_0} \left( \gamma^{(\tau)} \sum_{n=1}^{N} \left( M_{l,n} \odot \frac{\partial \mathcal{L}}{\partial W_{l,n}^{(\tau)}} \right) + \zeta^{(\tau)} \sum_{n=1}^{N} \left( M_{l,n} \odot W_{l,n}^{(\tau)} \right) \right)$$

$$= W_l^{(t_0)} - \lambda^{(t_0)} \sum_{\tau=0}^{t_0} \left( \gamma^{(\tau)} \sum_{n=1}^{N} \left( M_{l,n} \odot \frac{\partial \mathcal{L}}{\partial W_{l,n}^{(\tau)}} \right) + \zeta^{(\tau)} W_l^{(\tau)} \right) \qquad \text{Equation 9}$$

We then show that when equation 9 holds for $t$ then:

$$\frac{\partial \mathcal{L}}{\partial W_{l,n}^{(t)}} = \frac{\partial \mathcal{L}}{\partial W_l^{(t)}}$$

First we demonstrate that the convolution gradient is only a function of the input tensor, $X^{(t)}$ and the output gradient, $\partial \mathcal{L} / \partial Y^{(t)}$, and not of the weight $W$. We make use of the tensor index and Einstein summation notation:

$$Y^{(t)} = W^{(t)} * X^{(t)}$$

$$Y(c_o, h, w)^{(t)} = \sum_{c_i=0}^{C_i-1} \sum_{k_h=0}^{K_H-1} \sum_{k_w=0}^{K_W-1} W(c_o, c_i, k_h, k_w)^{(t)} X(c_i, h+k_h, w+k_w)^{(t)}$$

$$Y_{c_o,h,w}^{(t)} = W_{c_o,c_i,k_h,k_w}^{(t)} X_{c_i,h,k_h,w,k_w}^{(t)}$$

$$\frac{\partial \mathcal{L}}{\partial W_{c_o,c_i,k_h,k_w}^{(t)}} = \sum_{k,i,j} \frac{\partial \mathcal{L}}{\partial Y_{k,i,j}^{(t)}} \frac{\partial Y_{k,i,j}^{(t)}}{\partial W_{c_o,c_i,k_h,k_w}^{(t)}}$$

$$= \frac{\partial \mathcal{L}}{\partial Y_{k,i,j}^{(t)}} \frac{\partial Y_{k,i,j}^{(t)}}{\partial W_{c_o,c_i,k_h,k_w}^{(t)}}$$

$$= \frac{\partial \mathcal{L}}{\partial Y_{k,i,j}^{(t)}} \frac{\partial W_{k,l,m,n}^{(t)}}{\partial W_{c_o,c_i,k_h,k_w}^{(t)}} X_{l,i,m,j,n}^{(t)}$$

$$= \frac{\partial \mathcal{L}}{\partial Y_{k,i,j}^{(t)}} \delta_{k,c_o} \delta_{l,c_i} \delta_{m,k_h} \delta_{n,k_w} X_{l,i,m,j,n}^{(t)}$$

$$= \frac{\partial \mathcal{L}}{\partial Y_{c_o,i,j}^{(t)}} X_{c_i,i,k_h,j,k_w}^{(t)}$$

$$= \sum_{i,j} \frac{\partial \mathcal{L}}{\partial Y_{c_o,i,j}^{(t)}} X_{c_i,i,k_h,j,k_w}^{(t)} \quad \Rightarrow \quad \frac{\partial \mathcal{L}}{\partial W^{(t)}} = g \left( \frac{\partial \mathcal{L}}{dY^{(t)}}, X^{(t)} \right)$$

Given equation 9 we can now show:

$$Y^{(t_0)} = \sum_{n=1}^{N} \left( \left( M_n \odot W_n^{(t_0)} \right) * X^{(t_0)} \right) = \sum_{n=1}^{N} Y_n^{(t_0)} \quad \Rightarrow \quad \frac{\partial \mathcal{L}}{\partial Y_n^{(t_0)}} = \frac{\partial \mathcal{L}}{\partial Y^{(t_0)}}$$

which implies that:

$$\frac{\partial \mathcal{L}}{\partial W_{l,n}^{(t)}} = g\left( \frac{\partial \mathcal{L}}{dY_n^{(t)}}, X^{(t)} \right) = g\left( \frac{\partial \mathcal{L}}{dY^{(t)}}, X^{(t)} \right) = \frac{\partial \mathcal{L}}{\partial W_l^{(t)}}$$

Finally, picking up where we left off:

$$W_l^{(t_0+1)} \Leftarrow W_l^{(t_0)} - \lambda^{(t_0)} \sum_{\tau=0}^{t_0} \left( \gamma^{(\tau)} \sum_{n=1}^{N} \left( M_{l,n} \odot \frac{\partial \mathcal{L}}{\partial W_{l,n}^{(\tau)}} \right) + \zeta^{(\tau)} W_l^{(\tau)} \right)$$

$$= W_l^{(t_0)} - \lambda^{(t_0)} \sum_{\tau=0}^{t_0} \left( \gamma^{(\tau)} \sum_{n=1}^{N} \left( M_{l,n} \odot \frac{\partial \mathcal{L}}{\partial W_l^{(\tau)}} \right) + \zeta^{(\tau)} W_l^{(\tau)} \right)$$

$$= W_l^{(t_0)} - \lambda^{(t_0)} \sum_{\tau=0}^{t_0} \left( \gamma^{(\tau)} \sum_{n=1}^{N} \left( M_{l,n} \right) \odot \frac{\partial \mathcal{L}}{\partial W_l^{(\tau)}} + \zeta^{(\tau)} W_l^{(\tau)} \right)$$

$$= W_l^{(t_0)} - \lambda^{(t_0)} \sum_{\tau=0}^{t_0} \left( \gamma^{(\tau)} G_l \odot \frac{\partial \mathcal{L}}{\partial W_l^{(\tau)}} + \zeta^{(\tau)} W_l^{(\tau)} \right)$$

We arrive at lemma 1.

$$\begin{bmatrix} K_{-1,-1} & K_{-1,\,0} & K_{-1,+1} \\ K_{\,\,0,-1} & K_{\,\,0,\,0} & K_{\,\,0,+1} \\ K_{+1,-1} & K_{+1,\,0} & K_{+1,+1} \end{bmatrix} := \begin{bmatrix} 1/\beta & 1/\alpha & 1/\beta \\ 1/\alpha & 1 & 1/\alpha \\ 1/\beta & 1/\alpha & 1/\beta \end{bmatrix} \times \frac{9}{1 + 4/a + 4/b}$$

Figure 7: $\alpha$ and $\beta$ parameterization of the spatial gradient scaling used for the scaling grid search. $\alpha$ and $\beta$ determine the ratio between the center element and the edges and corners respectively. An additional multiplication factor is added to ensure a normalized mean.

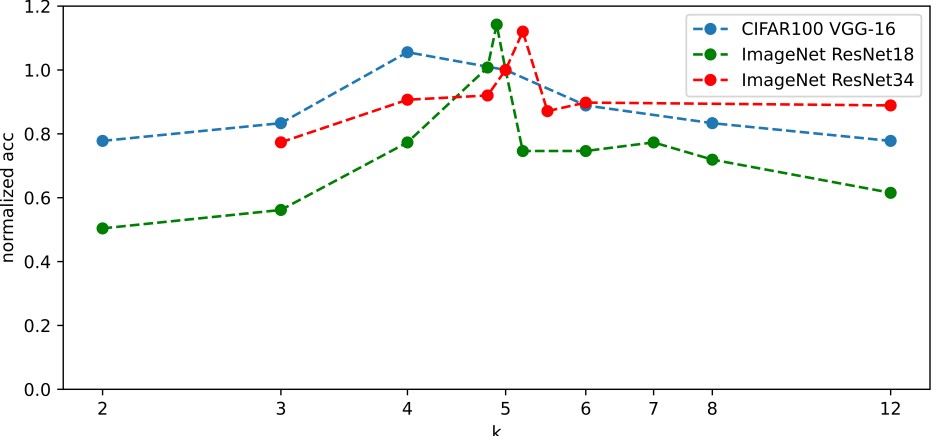

Figure 8: Results for K search for VGG-16 on CIFAR100 and ResNets on ImageNet. Accuracies are normalized such that the baseline is 0 and reported results of $k = 5$ is 1.

