# OpenReview forum: "Reparameterization through Spatial Gradient Scaling"
_ICLR.cc/2023/Conference — ICLR 2023 poster_

### Official Review · Reviewer_xBEc · 2022-10-24

**Confidence:** 3
**Correctness:** 4
**Technical Novelty And Significance:** 3
**Empirical Novelty And Significance:** 2
**Recommendation:** 6

**Clarity, Quality, Novelty And Reproducibility:**

**Clarity:**

The paper is well-paced with good explanations and illustrations. The paper contains sufficient technical details.

**Quality:**

I am convinced that their proposed spatial gradient scaling is equivalent to branched reparameterization in previous work without introducing structural changes. I have no questions regarding this contribution and I think their experiments also support that claim well.

However, I do not understand why structural reparameterization is an important and promising direction. As shown by this paper, previous work on structural reparameterization can just be seen as gradient rescaling. Can this line of research be seen as designing data-dependent optimizers dedicated to convolutional neural networks? If that is the case, can we find other optimization methods with adaptive gradients such that gradient rescaling is no longer essential? I think the baselines should not be previous work on structural reparameterization, because we already know they are equivalent. The paper should try to demonstrate why structural reparameterization/spatial gradient scaling is an attractive technique and the practitioners should be recommended to use it. If this cannot be justified theoretically, then more empirical studies need to be conducted to show that performance improvement is robust and can be used on a variety of tasks/architectures, and no other techniques (e.g. adaptive gradient optimizers) can not achieve similar improvement.

**Reproducibility:**

The paper provided enough experimental details. The code is currently not provided.

**Details Of Ethics Concerns:**

I have no echical concerns.

**Strength And Weaknesses:**

**Strength:**
1. This paper is relatively well-written and most technical parts are clear to me. The experimental details are also sufficient.
2. The connection between spatial gradient scaling and parallel convolution parameterization is a good observation. It shows that previous work on muti-branch reparameterization is unnecessary for achieving better performance.

**Weaknesses:**
1. The proposed approach is heuristic and lacks theoretical foundations. The use of mutual information for gradient rescaling is intuitive but it is hard to justify from a theoretical perspective.
2. There are useful tricks in deep learning which do not have a theoretical explanation in the beginning. But in that case, the empirical study needs to be more convincing. It may not be enough to only apply their technique to classification tasks and compared only to previous work on structural re-parameterization. In my opinion, it would be convincing if they could show that their performance improvement can not be possibly achieved by using other optimization methods with adaptive gradient (e.g. AdaGrad), and their empirical gain is robust across tasks such as object recognition, image segmentation, and image generation, and across a wide range of base CNN architectures. Of course, I am not asking the authors to run these experiments during the rebuttal period. These are just suggestions to make a strong paper in the future. That level of empirical study is usually not necessary, but for a more heuristic technique, it might be good to demonstrate the robustness of the performance gain.

**Summary Of The Paper:**

This paper proposes to rescale gradient components based on spatial mutual information when optimizing convolutional neural networks. They show that the proposed spatial gradient scaling is equivariant to previous work on structural re-parameterization (which uses a multi-branch topology network during training and re-parameterize back to the original convolution for inference). They compare their approach to previous work using structural re-parameterization for classification tasks on the CIFAR and ImageNet datasets and demonstrate that the proposed technique improves performance at a lower computational cost.


**Summary Of The Review:**

I am leaning toward weak rejection but I think this paper still has merits.

Their claim that the proposed spatial gradient scaling is equivalent to branched reparameterization in previous work without introducing structural changes is correct and it is also supported by their experiments.

However, the proposed technique is too heuristic with weak theoretical justification. In experiments, they only compare to branched reparameterization in previous work  (which has already been shown to be equivalent to their proposed approach). So I am not convinced that this technique can bring robust performance gains that cannot be achieved using other optimization algorithms with adaptive gradients.

---

> ### Author Response · Authors · 2022-11-18
> **Author response to Reviewer xBEc (part 1/4)**
>
> ## 1. Comparisons to Adaptive Gradient Optimizers
>
>  The proposed spatial gradient scaling is different from prior work. Unlike Adagrad, which scales gradients before model update based on model weights and gradients of previous timesteps, the proposed SGS performs gradient scaling spatially among different weights in a kernel based on the spatial dependencies within data. To further showcase the difference between methods, we have added experimental comparisons between our spatial gradient scaling and various adaptive optimizers, including Adagrad, as shown below. Details of our method is given in Appendix A.2.
>
> | **Model**     | **Optimizer**  | **No SGS (%)**  | **With SGS (%)**  |
> |---------------|----------------|-----------------|---------------|
> | VGG-11        | SGD            | 68.3±0.1        | 70.7±0.1      |
> |               | SGD + Momentum | 71.0±0.1        | 73.2±0.1      |
> |               | Adadelta       | 69.2±0.2        | 70.8±0.1      |
> |               | Adagrad        | 66.2±0.1        | 66.2±0.2      |
> |               | Adagrad\*       | 66.2±0.1        | 67.0±0.1      |
> |               | Adam           | 68.8±0.2        | 70.4±0.2      |
> |               | Adamax         | 68.4±0.2        | 69.9±0.1      |
> |               | NAdam          | 69.2±0.1        | 70.2±0.1      |
> |               | RAdam          | 69.8±0.1        | 71.1±0.1      |
> |               |                |                 |               |
> | ResNet-18     | SGD            | 75.8±0.2        | 76.3±0.1      |
> |               | SGD + Momentum | 77.6±0.1        | 78.5±0.1      |
> |               | Adadelta       | 75.8±0.2        | 76.4±0.1      |
> |               | Adagrad        | 72.2±0.2        | 72.3±0.2      |
> |               | Adagrad\*       | 72.2±0.2        | 72.9±0.1      |
> |               | Adam           | 75.1±0.1        | 76.2±0.1      |
> |               | Adamax         | 74.9±0.1        | 75.8±0.1      |
> |               | NAdam          | 75.6±0.1        | 76.0±0.1      |
> |               | RAdam          | 76.2±0.2        | 76.8±0.1      |
>
>  First, we can see from the table that SGD + Momentum outperforms all other optimizers by significant margins. This is why we have used SGD + Momentum (which also incorporates historical gradients) as the optimizer to generate our experimental results. These optimizer comparison results also agree with the results reported by Wilson2017TheMV[1].
>
> Furthermore, we find that if spatial gradient scaling is further applied on top of an optimizer, there is performance gain for most optimizers, verifying the robustness and applicability of SGS to a wide range of optimizers, the only exception being Adagrad. The reason is that when applying gradient scaling to an optimizer, we can either apply the scaling directly to the backpropagation gradients (before optimizer calculations) or after optimizer calculations (before the update step). We usually perform the former option (lines with no \*). However, we find that by doing the latter option, we can also achieve performance gains on Adagrad (shown as Adagrad\* in the table). Finally, as seen in the table, applying SGS to SGD + Momentum yeilds a significant performance gain, largely outperforming other optimizer baselines. These results show that the performance improvement of our proposed spatial gradient scaling is distinguishing and is not achievable by previously proposed adaptive gradient optimizers alone.
>
> [1] Wilson, Ashia C. et al. “The Marginal Value of Adaptive Gradient Methods in Machine Learning.” ArXiv abs/1705.08292 (2017): n. pag.

---

> > ### Author Response · Authors · 2022-11-18
> > **Author response to Reviewer xBEc (part 2/4)**
> >
> > ## 2. Performance Gains on Other CV Tasks
> >
> > To show the versatility of spatial gradient scaling, we look at its use in the context of other tasks. We consider Semantic Segmentation and Human Pose Estimation. Spatial gradient scaling can be similarly applied to other CV tasks as it is applied to image classification, i.e., by calculating the proposed mutual information metric for each tensor and scaling the gradients in a kernel spatially, as detailed in the paper (Section 3).
> >
> > For Semantic Segmentation, we train and evaluate on the Cityscapes dataset the PSPNet models, which are based on ResNet-18/34/50 backbones and are implemented by SemSeg [1]. Results are shown in the table below:
> >
> > | **Model**     | **Baseline (mIoU/mAcc/allAcc)**   | **SGS (mIoU/mAcc/allAcc)**        |
> > |---------------|-----------------------------------|------------------------------------|
> > | PSPNet-18     | 72.8/80.0/95.3                    | 73.7/81.1/95.4                     |
> > | PSPNet-34     | 76.1/84.0/95.7                    | 76.8/84.3/95.9                     |
> > | PSPNet-50     | 75.4/82.8/95.8                    | 75.6/82.8/95.9                     |
> >
> > Training details are given in Appendix A.4. We find performance improvements across all models.
> >
> > For Human Pose Estimation, we train and evaluate on MPII dataset with MSRA ResNet using the training code of Pytorch-Pose-HG-3D [2]. MSRA ResNet also shares its backbone with the ResNet family. The results for Human Pose Estimation are as follows:
> >
> > | **Model**          | **Baseline (PCKh\@0.5)** | **SGS (PCKh\@0.5)**  |
> > |--------------------|-------------------------|----------------------|
> > | MSRA ResNet-18     | 83.8                    | 84.4                 |
> > | MSRA ResNet-34     | 85.8                    | 86.2                 |
> > | MSRA ResNet-50     | 85.3                    | 85.6                 |
> >
> > We again see performance gains across models on HPE. Overall, although image classification lends itself to benchmarking, SGS is not limited to image classification tasks, but can also be used to improve learning performance of other CV tasks involving convolutional networks.
> >
> > [1] https://github.com/hszhao/semseg
> >
> > [2] https://github.com/xingyizhou/pytorch-pose-hg-3d

---

> > > ### Author Response · Authors · 2022-11-18
> > > **Author response to Reviewer xBEc (part 3/4)**
> > >
> > > ## 3. Performance Gains Across CNN Architectures
> > > We show that spatial gradient scaling can benefit the training of diverse types of CNN architectures. We have added the following experimental results:
> > >
> > > | **Model**              | **Baseline (%)** | **With SGS (Ours) (%)**      |
> > > |------------------------|------------------|-------------------|
> > > | VGG-11                 | 70.8±0.1         | 73.2±0.1          |
> > > | VGG-13                 | 74.7±0.1         | 76.1±0.1          |
> > > | VGG-16                 | 74.9±0.1         | 76.2±0.1          |
> > > | VGG-19                 | 74.6±0.1         | 76.1±0.2          |
> > > | ResNet-18              | 77.8±0.1         | 78.8±0.1          |
> > > | ResNet-34              | 79.0±0.3         | 80.0±0.2          |
> > > | ResNet-50              | 79.1±0.2         | 79.6±0.2          |
> > > | ShuffleNet             | 72.4±0.1         | 72.8±0.2          |
> > > | MobileNetV2            | 73.9±0.1         | 74.2±0.2          |
> > > | SeResNet-18            | 78.4±0.1         | 79.6±0.1          |
> > > | SeResNet-34            | 79.3±0.2         | 80.3±0.2          |
> > > | PreActResNet-18        | 75.7±0.1         | 76.3±0.1          |
> > > | PreActResNet-34        | 77.4±0.1         | 77.9±0.1          |
> > > | PyramidNet (α=48)      | 79.1±0.1         | 79.9±0.2          |
> > > | PyramidNet (α=84)      | 81.1±0.1         | 81.6±0.1          |
> > > | PyramidNet (α=270)     | 83.3±0.1         | 83.6±0.1          |
> > > | Xception               | 79.0±0.1         | 80.0±0.1          |
> > >
> > > Details of these experiments are given in Appendix A.1. We observe that SGS yields performance gains for a range of models beyond VGG and ResNets.

---

> > > > ### Author Response · Authors · 2022-11-18
> > > > **Author response to Reviewer xBEc (part 4/4)**
> > > >
> > > > ## 4. Robust Performance Across Training Settings
> > > >
> > > > We further conduct experiments to demonstrate that the gain from SGS is not sensitive to some training hyperparameters. The results are shown below and in Appendix A.3:
> > > >
> > > > | **Model**     | **Epochs** | **Batch Size** | **Baseline (%)** | **With SGS (Ours) (%)**     |
> > > > |---------------|------------|----------------|------------------|------------------|
> > > > | VGG-11        | 100        | 64             | 70.4±0.2         | 71.1±0.1         |
> > > > |               |            | 128            | 70.8±0.1         | 72.5±0.2         |
> > > > |               |            | 512            | 68.8±0.2         | 70.8±0.1         |
> > > > |               | 200        | 64             | 71.0±0.1         | 72.4±0.2         |
> > > > |               |            | 128            | 70.8±0.1         | 73.2±0.1         |
> > > > |               |            | 512            | 68.6±0.2         | 71.6±0.1         |
> > > > |               | 600        | 64             | 71.0±0.1         | 73.0±0.1         |
> > > > |               |            | 128            | 70.1±0.2         | 73.2±0.1         |
> > > > |               |            | 512            | 68.2±0.1         | 71.3±0.1         |
> > > > |               |            |                |                  |                  |
> > > > | ResNet-18     | 100        | 64             | 77.5±0.2         | 78.1±0.1         |
> > > > |               |            | 128            | 77.5±0.1         | 78.0±0.2         |
> > > > |               |            | 512            | 75.2±0.1         | 75.6±0.1         |
> > > > |               | 200        | 64             | 77.8±0.2         | 78.4±0.1         |
> > > > |               |            | 128            | 77.8±0.1         | 78.8±0.1         |
> > > > |               |            | 512            | 76.2±0.1         | 76.7±0.1         |
> > > > |               | 600        | 64             | 78.0±0.1         | 78.7±0.1         |
> > > > |               |            | 128            | 77.9±0.1         | 79.0±0.1         |
> > > > |               |            | 512            | 76.4±0.1         | 77.2±0.2         |
> > > >
> > > > We find that our approach offers performance gains over the baseline under all tested configurations of epochs and batch sizes. This suggests that SGS can be used effectively to improve training and convergence in deep learning without requiring extensive hyperparameter tweaking.

---

> > > > > ### Comment · Reviewer_xBEc · 2022-11-21
> > > > > **Response to Authors**
> > > > >
> > > > > Although the proposed technique is heuristic and it is hard to justify their choices from a theoretical perspective. But during the rebuttal periods, the authors have run extensive experiments showing that their proposed technique brings robust performance gains across different tasks, architectures, and training settings. Even though the performance gains are generally small, there are consistent improvements in all scenarios. It proves that this technique is indeed valuable in practice. I would like to raise my score. In the future, it might be better to have a deeper understanding of the proposed method.

---

### Official Review · Reviewer_v3Lx · 2022-10-25

**Confidence:** 4
**Correctness:** 3
**Technical Novelty And Significance:** 4
**Empirical Novelty And Significance:** 4
**Recommendation:** 8

**Clarity, Quality, Novelty And Reproducibility:**

The paper is clear, the idea is good and overall the paper is of high quality. Reproducibility might be a concern so I would encourage the authors to release the code.

**Strength And Weaknesses:**

## Strengths
1. The idea is simple and elegant. It replaces structural reparametrizations which require high memory and computational overhead during training with a simple gradient scaling that performs similar to such methods.
2. The paper is well-written, and the idea is backed by the proofs and experimental results.

## Weaknesses
1. It is not clear why mutual information is a good measure to obtain gradient scales. Specifically, it is not clear to me why one should weight a filter coefficient higher if it is similar to the center pixel in a convolution filter. Even though the results show the benefits it would be good to motivate it intuitively. Please comment on this.
2. Even though the idea is simple, there were several implementation details (eg., the last para on page 6) required to make it work in practice. I think reproducibility might be a concern due to this.

**Summary Of The Paper:**

The paper provides a gradient scaling approach to replicate structural reparametrization that improves training. The main idea is based on the linearity of convolutions and the observation that the effect of reparametrization can be achieved by simple gradient scaling. The results show the benefits of the approach on multiple vision datasets.

**Summary Of The Review:**

Overall, the idea is nice and it is backed by theory and experiments.

## Post rebuttal
I thank the authors for the response. I would recommend adding a paragraph about the intuition behind using mutual information for gradient scaling. Additionally, the extensive experiments across various settings show the practical benefits of the method. I was already positive and I would like to see the paper accepted.

---

> ### Author Response · Authors · 2022-11-18
> **Author response to Reviewer v3Lx (part 1/2)**
>
> Thanks for reading our paper and for the valuable comments.
>
> The intuition of SGS is that 1) different weights in a convolutional kernel should be trained at different rates (by scaling gradients spatially); 2) how concentrated (toward the center of the kernel) the SGS is should depend on the spatial dependencies of neighboring pixels in images of the training data. The rationales are the following:
>
> First, the spatial relative importance of different weights in a kernel was first pointed out by Ding2019ACNetST, which shows through pruning experiments, that pruning the center and edge weights of a convolutional filter causes more harm to the classification accuracy than pruning corner weights (refer to pg.7, Figure 5 of Ding2019ACNetST) [1]. That is, they conclude that the relative importance of weights within a convolution kernel is determined by the spatial positions of these weights (center/edge/corner). In SGS, we leverage this observation to prioritize the learning of center weights over edge/corner weights.
>
> Second, we take one step further to numerically assess the relative importance of each kernel weight using the notion of spatial dependence of neighboring pixels across the image. This is because convolution kernels aim to detect features (local patterns) in an input image. For example, if patterns are non-existent, i.e., when the image is synthesized by random pixels spread out spatially, the neighboring pixels are independent and uncorrelated, there would be no local pattern for a convolution kernel of size 3x3 to detect. In this case, the spatial aggregation of a conv3x3 is wasted because no neighboring pixel interactions need to be detected by this conv3x3. In other words, structures and patterns in real-world natural images lead to the spatial correlation between pixels, and thus can be detected by convolutional kernels applied layer after layer. However, different images have different levels of structures/patterns in them, corresponding to different degrees of dependencies between pixels. We quantitively characterize such dependence for an image by comparing distributions of spatially separated pixels within a fixed-sized receptive field (e.g., 3x3) sweeped through the image to get a mutual information measure. This per-image measure is further aggregated over multiple batches of data to derive an overall dependence measure, which is used to determine how concentrated or flat the spatial gradient scaling should be during training.
>
> For example, for small resolution datasets like CIFAR (in comparison to ImageNet), a grid search of the optimal gradient scaling scheme (that leads to higher accuracy) would assign very low importance to the corner and edge elements, whereas for ImageNet, the optimal scaling given by grid search is relatively more spatially flat because intuitively, neighboring pixels will have higher dependences within a same kernel of 3x3 as image resolution increases. This is also reflected by the proposed mutual information-based SGS, which observes lower dependencies between neighboring pixels for images in CIFAR than higher-resolution images in ImageNet (when the same 3x3 kernel is applied), and thus uses more centered SGS for CIFAR and yet more flat SGS for ImageNet. Generally, the relative spatial importance assigned by mutual information agrees in trend with the results for optimal gradient scaling obtained from grid search (the latter is more costly and has worse performance).
>
> The competence of mutual information as a correlation measure is supported by an ablation study on correlation functions and a comparison to the gradient scaling grid search we do in Section 4.3 of the paper.
>
> [1] Ding, Xiaohan et al. “ACNet: Strengthening the Kernel Skeletons for Powerful CNN via Asymmetric Convolution Blocks.” 2019 IEEE/CVF International Conference on Computer Vision (ICCV) (2019): 1911-1920.

---

> > ### Author Response · Authors · 2022-11-18
> > **Author response to Reviewer v3Lx (part 2/2)**
> >
> > To clarify how the scaling is derived in SGS, we first calculate the mutual information between every pixel and its neighbors (see Figure 3), which are all the pixels within the convolutional receptive field at question (e.g. 3x3, 5x5). For example, a conv3x3 has a receptive field equivalent to the one-hop neighborhood of each pixel. Equation 7 gives the mutual information between each pixel in the feature map of layer l and its neighbor with displacement (i,j), based on their marginal and joint probability distributions. The marginal distribution of individual pixels can be statistically estimated via a histogram of pixel values in the image. Similarly, the joint distribution of each pixel and its (i,j)-neighbor can be estimated via a 2-D histogram of pixel pairs. Finally, we translate the estimated mutual information to actual scaling factors in SGS using the nonlinear function according to Equation 8, where the hyperparameter k is found using grid search on CIFAR100 (as shown in Figure 8, k=5) and works on a diverse range of models and datasets including newly added results. Such details of experimental settings can be found in Section 4.1 (CIFAR), 4.2 (ImageNet), and 4.3 (Ablation Studies).
> >
> > We have also added pseudo-code in Appendix A.5 regarding our SGS implementation and will polish the presentation.

---

### Official Review · Reviewer_ahFw · 2022-10-25

**Confidence:** 4
**Correctness:** 4
**Technical Novelty And Significance:** 3
**Empirical Novelty And Significance:** 3
**Recommendation:** 6

**Clarity, Quality, Novelty And Reproducibility:**

Paper is written clearly with graphs that help understand the methodology.
Reproducibility: I did not find a link to the code

**Strength And Weaknesses:**

Strengths:
- The paper seems novel.
- Derivations seem correct

Weaknesses:
- Figures seem to be in low resolution
- Table 1 evaluation is only with VGG (quite old), not sure why it did not include ResNet or more recent networks.
- Results on CIFAR 100 are quite low compared to the state-of-the-art
- I missed a comparison with other reparameterization methods, e.g. Salimans and Kingma (Neurips, 2016) and others.

**Summary Of The Paper:**

This paper proposes a method to scale the gradients in convolutions through learning pixel/feature dependencies

**Summary Of The Review:**

Paper seems interesting, although there are a few improvements that would benefit the paper

---

> ### Author Response · Authors · 2022-11-18
> **Author response to Reviewer ahFw**
>
> Thank you for taking the time to read and review our paper. We have updated our figures to be higher resolution.
>
> Following your recommendation, we have extended our CIFAR-100 results to include ResNet models and more recent networks with a diverse range of accuracies. The table can be found in Appendix A.1 as well as below:
>
> | **Model**              | **Baseline (%)** | **With SGS (Ours) (%)**|
> |------------------------|------------------|------------------------|
> | VGG-11                 | 70.8±0.1         | 73.2±0.1               |
> | VGG-13                 | 74.7±0.1         | 76.1±0.1               |
> | VGG-16                 | 74.9±0.1         | 76.2±0.1               |
> | VGG-19                 | 74.6±0.1         | 76.1±0.2               |
> | ResNet-18              | 77.8±0.1         | 78.8±0.1               |
> | ResNet-34              | 79.0±0.3         | 80.0±0.2               |
> | ResNet-50              | 79.1±0.2         | 79.6±0.2               |
> | ShuffleNet             | 72.4±0.1         | 72.8±0.2               |
> | MobileNetV2            | 73.9±0.1         | 74.2±0.2               |
> | SeResNet-18            | 78.4±0.1         | 79.6±0.1               |
> | SeResNet-34            | 79.3±0.2         | 80.3±0.2               |
> | PreActResNet-18        | 75.7±0.1         | 76.3±0.1               |
> | PreActResNet-34        | 77.4±0.1         | 77.9±0.1               |
> | PyramidNet (α=48)      | 79.1±0.1         | 79.9±0.2               |
> | PyramidNet (α=84)      | 81.1±0.1         | 81.6±0.1               |
> | PyramidNet (α=270)     | 83.3±0.1         | 83.6±0.1               |
> | Xception               | 79.0±0.1         | 80.0±0.1               |
>
> We see that using spatial gradient scaling can improve the performance of a wide range of models.
>
> We have also conducted experiments to compare to more of other reparameterization methods as suggested by the review. The results are shown below:
>
> | **Model** | **Rep method**   | **Cost (GPU hrs.)** | **Acc (%)** |
> |-----------|---------------------|---------------------|--------------|
> | VGG-11    | Baseline            | 0.75                | 70.8±0.2     |
> |           | Ours                | 0.78                | **73.2±0.1** |
> |           | Weight Normalization| 0.77                | 71.1±0.2     |
> |           | ACNet               | 1.22                | 72.6±0.1     |
> |           | DBB                 | 1.51                | 72.8±0.1     |
> |           | DyRep               | 1.17                | 71.5±0.2     |
> |           |                     |                     |              |
> | ResNet-18 | Baseline            | 1.08                | 77.8±0.1     |
> |           | Ours                | 1.18                |**78.8±0.1**  |
> |           | Weight Normalization| 1.1                 | 78.0±0.1     |
> |           | ACNet               | 2.33                | 78.4±0.1     |
> |           | DBB                 | 3.69                | 77.8±0.1     |
> |           | DyRep               | 1.67                | 77.9±0.1     |
>
> We see that our spatial gradient scaling outperforms other reparameterization methods at a low computational cost. Interestingly, we also find that these approaches may not be in complete opposition, and greater benefits can be attained by combining two or more reparameterization techniques. For example, we see a synergy between our spatial gradient scaling (SGS) and weight normalization (WN), as shown below:
>
> | **VGG-11** | **Baseline** | **WN**   | **SGS**  | **SGS + WN**  |
> |------------|--------------|----------|----------|---------------|
> | **Acc (%)**| 70.8±0.2     | 71.1±0.2 | 73.2±0.1 | 73.5±0.1      |

---

### Official Review · Reviewer_SeDu · 2022-10-30

**Confidence:** 3
**Correctness:** 3
**Technical Novelty And Significance:** 3
**Empirical Novelty And Significance:** 4
**Recommendation:** 8

**Clarity, Quality, Novelty And Reproducibility:**

The text is well-written. The source code is not shared. However, it seems possible to reimplement the method myself.

**Strength And Weaknesses:**

The paper is well-written. It has a coherent story and it is interesting to read. The related work section is properly organized. As I was not aware of some of the papers mentioned there, I found it extremely useful.

I find the idea of the paper elegant. Although it is simple, it demonstrates that a simple modification of the learning process can achieve the same result (or better) than a modification of the architecture.

I didn't notice any significant issues. However, I would appreciate if the authors explain why they use mutual information for gradient rescaling. Why not some other function? And if it is not limited to mutual information, would the results differ if the method relies on some other measure for spatial correlation?

**Summary Of The Paper:**

The paper proposes to scale the components of the gradients during back-propagation to achieve the same results as branched reparametrization.

**Summary Of The Review:**

The paper proposes a novel and elegant idea. I think it is worth sharing with the community.

---

> ### Author Response · Authors · 2022-11-18
> **Author response to Reviewer SeDu**
>
> Thank you for your review and kind words. We're happy you enjoyed the paper.
>
> The choice to use mutual information for gradient scaling was motivated both empirically and intuitively. In the paper, we discuss a comparison to autocorrelation (Sec 4.3), where we show that mutual information outperforms both autocorrelation and a gradient scaling grid search. We show the table of results below:
>
> | **SGS Method**       | **Cost (GPU hrs.)**| **Acc (%)**  |
> |----------------------|--------------------|--------------|
> | Baseline             | 0.40               | 69.6±0.1     |
> | Grid Search          | 19.6               | 71.0±0.1     |
> | Autocorrelation      | 0.42               | 71.1±0.2     |
> | Mutual Information   | 0.42               | 71.5±0.1     |
>
> We have also run additional experiments using other correlation measures (Pearson, etc.). While all these correlation measures improved baseline performance, mutual information performed the best. We attribute mutual information's superior performance to its ability to capture both linear and non-linear dependencies.

---

### Author Response · Authors · 2022-11-18
**Summary of Changes**

Thank you for all the reviews. Based on the comments, we have made the following revision to the paper:
* Added CIFAR-100 experiments on more types of CNNs to showcase the benefit and general applicability of spatial gradient scaling (SGS) on a diverse range of convolutional networks (Appendix A.1).
* Comparison to adaptive gradient optimizers (Adagrad, Adam, etc.) (Appendix A.2). We show our approach yields unique performance gains unattainable by existing adaptive gradient optimizers. We also show that SGS benefits a range of optimizers when applied on top of them.
* Added a study to examine the sensitivity of the performance gains of SGS to training hyperparameters (epochs and batch size) (Appendix A.3). We find performance gains with SGS under all tested configurations.
* We ran additional experiments on other CV tasks, including semantic segmentation and human pose estimation (Appendix A.4). We find that training on these tasks can also benefit from the use of spatial gradient scaling.
* Added pseudo-code for model training with SGS in Appendix A.5.
* Rewrite parts of the Introduction and Related Work for clarity.

---

### Decision · Program_Chairs · 2023-01-20

**Decision:**

Accept: poster

**Justification For Why Not Higher Score:**

Because the reported empirical gains are not significant and there is no theoretical justification for the proposed method.

**Justification For Why Not Lower Score:**

Because the proposed idea is novel and leads to empirical gains.

**Metareview: Summary, Strengths And Weaknesses:**

This paper proposes a method to rescale components of the gradient during the back propagation to achieve similar performance as branched reparametrization. The paper is well-written and easy to follow. The proposed approach is novel and leads to marginal improvements. Therefore, I recommend acceptance.

**Note From Pc:**

if the above contains the word "oral" or "spotlight" please see: "oral" presentation means -> notable-top-5% and "spotlight" means -> notable-top-25%. As stated in our emails, we are disassociating presentation type from AC recommendations